# Lefamulin in Patients with Community-Acquired Bacterial Pneumonia Caused by Atypical Respiratory Pathogens: Pooled Results from Two Phase 3 Trials

**DOI:** 10.3390/antibiotics10121489

**Published:** 2021-12-04

**Authors:** Susanne Paukner, David Mariano, Anita F. Das, Gregory J. Moran, Christian Sandrock, Ken B. Waites, Thomas M. File

**Affiliations:** 1Nabriva Therapeutics GmbH, Leberstrasse 20, 1110 Vienna, Austria; 2Nabriva Therapeutics US, Inc., Fort Washington, PA 19034, USA; David.Mariano@Nabriva.com; 3Das Consulting, Guerneville, CA 95446, USA; Adas@ADstat.onmicrosoft.com; 4Olive View-UCLA Medical Center, Los Angeles, CA 91342, USA; gmoran@ucla.edu; 5Department of Internal Medicine, UC Davis School of Medicine, Sacramento, CA 95817, USA; cesandrock@ucdavis.edu; 6Department of Pathology, University of Alabama at Birmingham, Birmingham, AL 35294, USA; waiteskb@yahoo.com; 7Summa Health, Akron, OH 44304, USA; filet@summahealth.org

**Keywords:** atypical pathogens, lefamulin, community-acquired bacterial pneumonia, *Mycoplasma pneumoniae*, *Chlamydia pneumoniae*, *Legionella pneumophila*

## Abstract

Lefamulin was the first systemic pleuromutilin antibiotic approved for intravenous and oral use in adults with community-acquired bacterial pneumonia based on two phase 3 trials (Lefamulin Evaluation Against Pneumonia [LEAP]-1 and LEAP-2). This pooled analysis evaluated lefamulin efficacy and safety in adults with community-acquired bacterial pneumonia caused by atypical pathogens (*Mycoplasma pneumoniae*, *Legionella pneumophila*, and *Chlamydia pneumoniae*). In LEAP-1, participants received intravenous lefamulin 150 mg every 12 h for 5–7 days or moxifloxacin 400 mg every 24 h for 7 days, with optional intravenous-to-oral switch. In LEAP-2, participants received oral lefamulin 600 mg every 12 h for 5 days or moxifloxacin 400 mg every 24 h for 7 days. Primary outcomes were early clinical response at 96 ± 24 h after first dose and investigator assessment of clinical response at test of cure (5–10 days after last dose). Atypical pathogens were identified in 25.0% (91/364) of lefamulin-treated patients and 25.2% (87/345) of moxifloxacin-treated patients; most were identified by ≥1 standard diagnostic modality (*M. pneumoniae* 71.2% [52/73]; *L.* *pneumophila* 96.9% [63/65]; *C. pneumoniae* 79.3% [46/58]); the most common standard diagnostic modality was serology. In terms of disease severity, more than 90% of patients had CURB-65 (confusion of new onset, blood urea nitrogen > 19 mg/dL, respiratory rate ≥ 30 breaths/min, blood pressure <90 mm Hg systolic or ≤60 mm Hg diastolic, and age ≥ 65 years) scores of 0–2; approximately 50% of patients had PORT (Pneumonia Outcomes Research Team) risk class of III, and the remaining patients were more likely to have PORT risk class of II or IV versus V. In patients with atypical pathogens, early clinical response (lefamulin 84.4–96.6%; moxifloxacin 90.3–96.8%) and investigator assessment of clinical response at test of cure (lefamulin 74.1–89.7%; moxifloxacin 74.2–97.1%) were high and similar between arms. Treatment-emergent adverse event rates were similar in the lefamulin (34.1% [31/91]) and moxifloxacin (32.2% [28/87]) groups. Limitations to this analysis include its post hoc nature, the small numbers of patients infected with atypical pathogens, the possibility of PCR-based diagnostic methods to identify non-etiologically relevant pathogens, and the possibility that these findings may not be generalizable to all patients. Lefamulin as short-course empiric monotherapy, including 5-day oral therapy, was well tolerated in adults with community-acquired bacterial pneumonia and demonstrated high clinical response rates against atypical pathogens.

## 1. Introduction

Pneumonia is associated with substantial morbidity, mortality, and economic burden [1,2,3] and is among the leading causes of infection-related deaths and hospitalizations in the United States [4,5]. Among adults with pneumonia, approximately 14% of infections worldwide are caused by the atypical pathogens *Mycoplasma pneumoniae*, *Chlamydia pneumoniae*, and *Legionella pneumophila* [6], and the proportion of pneumonia caused by these pathogens has increased over the last 15 years [7]. However, wide heterogeneity across countries has been observed in the estimated prevalence of atypical pathogens, which has been attributed to a lack of standardization in diagnostic testing [8]. Differences among countries include diagnostic approach, testing frequency, and a deficiency in widely available, specific, validated microbiologic tests [7,8]. This variance in testing for atypical pathogens may result in underdiagnoses and underreporting, which obscures the epidemiologic burden of atypical pathogens in pneumonia and could result in inappropriate antibiotic choice (e.g., beta-lactams) [6,7,8,9,10].

Pneumonia caused by atypical pathogens is typically mild to moderate in severity [10,11,12,13]. However, at least 25% of patients with atypical pathogens require hospitalization (associated with a 5.1% mortality rate), and nearly 1% are admitted to the intensive care unit (ICU) [10,14], primarily because of acute respiratory failure (associated with an 11% mortality rate) [12]. Of the atypical pathogens, *M. pneumoniae* is generally the most frequently isolated causative pathogen [7,14] and is associated with substantial morbidity and mortality, especially in the elderly [10]. An Israeli retrospective study of hospitalizations reported that nearly 40% of patients aged >65 years who tested positive for *M. pneumoniae* were admitted to the ICU, with a 46.4% mortality rate [10]. Although *L. pneumophila* is less frequently isolated compared with the other atypical pathogens [7,14], it is associated with the highest pneumonia severity and the quickest illness onset [7,11,15]. More-over, in the United States, a nearly 9-fold increase has been observed in cases of pneumonia caused by *L. pneumophila* between 2000 and 2018 [16].

The recommended antibiotics for treating community-acquired bacterial pneumonia (CABP) caused by atypical pathogens are macrolides and fluoroquinolones [17,18]. However, strains of *L. pneumophila* have been isolated with mutations that reduce its susceptibility to macrolides and to fluoroquinolones such as ciprofloxacin [19,20,21]. A worldwide emergence of macrolide-resistant *M. pneumoniae* has also been observed. Macrolide resistance rates of 2–20% have been reported in Europe [22,23,24,25], and rates as high as 92% have been observed in Asia [26]. Studies in the United States have identified macrolide resistance in 10–13% of *M. pneumoniae* samples overall [27,28], and a more recent US surveillance study reported macrolide resistance in 7.5% of *M. pneumoniae* specimens overall, with rates of resistant isolates in some regions exceeding 20% [29]. Increasing antibiotic resistance in atypical pathogens, as well as the adverse event profile of macrolides and fluoroquinolones, underscores the need for new antibiotics with novel mechanisms of action for the treatment of CABP [30,31,32].

Lefamulin, the first systemic pleuromutilin antimicrobial approved for intravenous (IV) and oral use in adults with CABP [33], inhibits bacterial protein synthesis and has demonstrated potent in vitro and in vivo activity against typical (e.g., *Streptococcus pneumoniae*, *Staphylococcus aureus*, *Haemophilus influenzae*) and atypical CABP pathogens, including those resistant to other major antibiotic classes [33,34,35,36,37,38]. Lefamulin has been shown to accumulate in macrophages at clinically relevant extracellular concentrations [38], which may explain its in vitro activity against intracellular pathogens such as *C. pneumoniae*, *L. pneumophila*, and *M. pneumoniae* [36,38,39]. In the phase 3 Lefamulin Evaluation Against Pneumonia (LEAP)-1 and LEAP-2 trials, lefamulin was noninferior to the standard of care, moxifloxacin, in adults with CABP [40,41]. In this pooled post hoc analysis of the LEAP-1 and LEAP-2 trials, we assessed the efficacy and safety of lefamulin versus moxifloxacin in adults with CABP caused by atypical respiratory pathogens.

## 2. Results

### 2.1. Patients

The pooled intent-to-treat (ITT) population (all randomized patients) included 1289 patients (lefamulin *n* = 646; moxifloxacin *n* = 643). Within the overall pooled microbiological ITT (microITT) population (lefamulin *n* = 364; moxifloxacin *n* = 345), atypical pathogens were identified in 91 patients (25.0%) treated with lefamulin and 87 (25.2%) treated with moxifloxacin. Patient demographics and baseline characteristics in this subgroup (Table 1) and in subgroups for each individual atypical pathogen (see Appendix A) were generally similar to those of the overall ITT population [42]. In terms of disease severity, more than 90% of patients had CURB-65 (confusion of new onset, blood urea nitrogen > 19 mg/dL, respiratory rate ≥ 30 breaths/min, blood pressure < 90 mm Hg systolic or ≤60 mm Hg diastolic, and age ≥ 65 years) scores of 0–2 (Table 1); approximately 50% of patients had PORT (Pneumonia Outcomes Research Team) risk class of III, and the remaining patients were more likely to have PORT risk class of II or IV versus V.

Of patients with *M*. *pneumoniae*, *L. pneumophila*, and *C. pneumoniae*, most (71.2% [52/73], 96.9% [63/65], and 79.3% [46/58], respectively) were identified by ≥1 standard diagnostic modality (i.e., culture, serology, or urinary antigen test), the most common of which was serology (Figure 1). At baseline, 98 patients (55.1%) had monomicrobial pneumonia, and 80 (44.9%) had polymicrobial pneumonia (Figure 2), of which coinfection with a Gram-positive pathogen (*S. pneumoniae* 29.8%; *S. aureus* 3.4%) was more frequent than with a Gram-negative pathogen (*H. influenzae* 11.8%; *M. catarrhalis* 4.5%) (Table 1). Minimum inhibitory concentration (MIC) values for *L. pneumophila* isolates collected from sputum (*n* = 2) were 0.5–1 µg/mL for lefamulin and 0.03 µg/mL for moxifloxacin. For *M*. *pneumoniae* isolates (*n* = 17), MIC values were ≤0.001 µg/mL for lefamulin, 0.125 µg/mL for moxifloxacin, 0.06–0.5 µg/mL for doxycycline, and ≤0.001 µg/mL for azithromycin. None of the laboratories were able to successfully culture *C. pneumoniae*.

### 2.2. Efficacy

Among patients with atypical pathogens at baseline, early clinical response (ECR) rates in the microITT and microITT-2 populations were high (lefamulin 84.4–96.6%, moxifloxacin 90.3–96.8%) and similar between treatment groups (Figure 3), consistent with ECR rates in the overall pooled microITT (lefamulin 89.3%; moxifloxacin 93.0%) and microITT-2 (lefamulin 90.0%; moxifloxacin 92.8%) populations. Patients with atypical pathogens at baseline also achieved high investigator assessment of clinical response (IACR) success rates at the test-of-cure (TOC) visit in the microITT and microITT-2 populations that were similar between treatment groups (Figure 3) and consistent with findings observed in the overall pooled populations. ECR and IACR success rates at TOC in the microITT population remained high regardless of whether patients had monomicrobial (ECR: lefamulin 90.0%, moxifloxacin 87.5%; IACR: lefamulin 76.5%, moxifloxacin 80.9%) or polymicrobial (ECR: lefamulin 90.0%, moxifloxacin 97.5%; IACR: lefamulin 90.0%, moxifloxacin 87.5%) infections. Among patients with atypical pathogens at baseline, microbiological response of success at TOC in the microITT population, which typically relied on clinical responses, was comparable between treatment groups and consistent with findings in the microITT-2 population (Figure 4).

### 2.3. Safety

Among patients with atypical pathogens at baseline, treatment-emergent adverse events (TEAE) rates were generally similar in the lefamulin (34.1% [31/91]) and moxifloxacin (32.2% [28/87]) groups (Table 2); most were mild or moderate in severity, with 4.5% of patients experiencing severe TEAEs. TEAEs rarely led to study drug discontinuation. All serious TEAEs were unrelated to treatment. Results were consistent with those observed in the overall pooled safety population and when reported by atypical pathogen (Appendix A). Among patients with atypical pathogens, TEAE system organ classes that occurred in >5% of patients in the lefamulin group were gastrointestinal disorders; infections and infestations; investigations; and respiratory, thoracic, and mediastinal disorders. The most frequently reported individual TEAEs were diarrhea (lefamulin *n* = 3 [3.3%]; moxifloxacin *n* = 2 [2.3%]) and nausea (*n* = 3 [3.3%]; *n* = 2 [2.3%]); of these events, most (70%) occurred in patients from the LEAP 2 study who received oral dosing.

## 3. Discussion

In patients with CABP due to atypical pathogens, oral and IV lefamulin as a short-course empiric monotherapy, including as a 5-day therapy, were well tolerated and associated with high clinical response rates (ECR, IACR success, and microbiological response of success). Efficacy and safety results in patients with atypical pathogens were similar in both populations analyzed (microITT and microITT-2) and when assessed by each atypical pathogen. The results were consistent with those of the overall study population, particularly among patients with atypical pathogens and medical history factors that often complicate disease management and may increase morbidity and mortality, including age ≥65 years or history of smoking, asthma/COPD, or diabetes [44,45,46].

Atypical pathogens are increasingly being recognized as a global public health problem [7,45]; however, testing for atypical pathogens in patients with CABP is not standardized, and widespread differences exist in testing frequency and diagnostic approach [8]. Even standard validated diagnostic assays, such as urine antigen testing for *Legionella* may not be routinely used [8]. In this post hoc pooled analysis of a subset of patients with CABP caused by atypical pathogens (*n* = 178), most atypical pathogens were identified by ≥1 standard diagnostic modality, and 45% of patients had polymicrobial pneumonia. However, in clinical practice, the use of multiple diagnostic modalities may not always be feasible (e.g., financial limitations) [8].

Difficulties in accurately identifying CABP-causing pathogens in the real-world setting and the presence of polymicrobial infections in adults with CABP underscore the importance of selecting an appropriate empirical antibiotic that effectively and safely treats both typical and atypical pathogens [47,48]. Evidence suggests that providing empiric antibiotic coverage for atypical pathogens may improve clinical outcomes and reduce economic burden. In a meta-analysis of five randomized controlled trials (*n* = 2011), the clinical failure rate among hospitalized patients with community-acquired pneumonia was significantly lower in patients who did versus did not receive such coverage (relative risk [95% confidence interval], 0.85 [0.73–0.99]; *p* = 0.037) [49]. Similarly, results of a multicenter, population-based, retrospective cohort study of 827 hospitalized patients with community-acquired pneumonia showed significant (all *p* < 0.01) benefits with respect to all-cause mortality in patients with (0.9%) versus without (4.9%) atypical coverage, as well as for mean length of stay (10.2 versus 11.6 days, respectively), total hospital cost (USD 1173 versus USD 1511, respectively), and direct antibiotic cost (USD 426 versus USD 503, respectively) [50]. Lefamulin has previously demonstrated potent in vitro activity against the most common typical and atypical CABP pathogens, including drug-resistant strains [34,35,36,51]. The current post hoc pooled analysis further demonstrates that lefamulin provides efficacy and safety generally similar to that of the respiratory fluoroquinolone, moxifloxacin, in patients with CABP caused by atypical pathogens.

This analysis was limited by the relatively low number of LEAP-1 and LEAP-2 patients with CABP caused by atypical respiratory pathogens (approximately 14% of the overall pooled study population), although this observation was generally consistent with previous estimates for atypical pathogens in patients with CABP [6]. A strength of this analysis was the use of a wide variety of diagnostic modalities, including standard detection methods such as serology, culture, and urine antigen testing as well as newer methodologies such as real-time qualitative polymerase chain reaction (RT-PCR), to ensure a sufficient population for analysis. The use of PCR-based diagnostic modalities has the potential to identify pathogens that are not etiologically or clinically relevant to a patient’s diagnosis. However, our results indicate that clinical response rates were high and similar between treatment groups regardless of whether the analysis population included (microITT population) or excluded (microITT-2 population) patients with baseline pathogens identified using PCR only. The LEAP-1 and LEAP-2 studies were not powered to detect statistically significant differences regarding atypical pathogens, and the results presented herein should be interpreted as exploratory descriptive analyses. Finally, these findings may not be generalizable to all patients with CABP caused by atypical pathogens, as the enrollment criteria for the LEAP-1 and LEAP-2 trials may have excluded some patients who would typically be seen in clinical practice. Most patients had CURB-65 scores of 0–2, reflective of mild to moderate disease, potentially limiting generalizability of the results to patients with more severe disease. However, approximately two-thirds of the patients had a PORT risk class of III or greater and one-quarter had multilobar pneumonia, suggesting that patients with severe pneumonia may have been reasonably represented by the study population.

## 4. Materials and Methods

Methods for the LEAP-1 (NCT02559310) and LEAP-2 (NCT02813694) multicenter, randomized, double-blind, double-dummy, phase 3 trials were previously described [40,41,42,52,53,54,55] and are briefly summarized here.

### 4.1. Study Design and Participants

In LEAP-1, patients were randomized (1:1) to receive IV lefamulin 150 mg every 12 h (q12h) or IV moxifloxacin 400 mg every 24 h (q24h; with alternating placebo doses to maintain blinding). Patients could switch to oral therapy (lefamulin 600 mg q12h or moxifloxacin 400 mg q24h) after 6 IV doses of study drug (approximately 3 days) if predefined improvement criteria were met. Treatment duration ranged from 5–10 days. In the initial study protocol, many patients received 5 days of lefamulin or 7 days of moxifloxacin, but patients with CABP due to *L. pneumophila* received 10 days of active treatment. A protocol amendment modified the treatment duration to 7 days for most patients in both groups, including those with CABP due to *L. pneumophila*. In LEAP-2, patients were randomized (1:1) to receive oral lefamulin 600 mg q12h for 5 days or oral moxifloxacin 400 mg q24h for 7 days (with matching oral placebo to maintain blinding).

Patients were included if they were aged ≥18 years with radiographically diagnosed pneumonia, PORT risk class III–V (LEAP-1, ≥25% PORT risk class IV/V) or II–IV (LEAP-2, ≥50% PORT risk class III/IV), acute onset of ≥3 CABP symptoms, ≥2 vital sign abnormalities, and ≥1 other clinical sign or laboratory finding of CABP. Exclusion criteria included ≥2 days of hospitalization within 90 days before symptom onset, receipt of >1 dose of a short-acting (dosing interval more frequent than q24h) oral or IV antibacterial for CABP within 72 h before randomization, severe immunosuppression, significant hepatic disease, creatinine clearance ≤ 30 mL/min, and being at risk of major cardiac events or dysfunction.

Before study initiation, centers obtained study approval from their respective ethics committees or institutional review boards [42]; all patients provided written informed consent. Trials were compliant with the ethical principles of the Declaration of Helsinki, Good Clinical Practice guidelines, and local laws and regulations.

### 4.2. Microbiological Assessments

Baseline atypical pathogens were identified from specimens collected within 24 h of the first dose of study drug. Diagnostic modalities varied by pathogen, and full details have been published previously [55]. Briefly, *M. pneumoniae* was identified by serology (≥4-fold increase in *M. pneumoniae* immunoglobulin [Ig] G serum antibody titer to ≥1:160 between baseline sample and convalescent sample collected at late follow-up visit (30 ± 3 days after first study drug dose) using *M. pneumoniae* antigen substrate slides and immunofluorescent antibody reagents [MBL Bion, Woburn, MA, USA]), culture from oropharyngeal specimens [56], and RT-PCR positive for the community-acquired respiratory distress syndrome toxin gene (*mpn372*) in sputum [57,58] or for the *repMp1* gene in oropharyngeal specimens [59,60]. *L. pneumophila* was identified by serology (≥4-fold increase in antibody titer to ≥1:128 by *L. pneumophila* group 1–6 indirect fluorescent antibody assay [Zeus Scientific, Branchburg, NJ, USA]), rapid urine antigen testing (BinaxNOW^®^; Legionella Urinary Antigen Card Abbott Diagnostics Scarborough, Inc., Scarborough, ME, USA), sputum culture, or RT-PCR positive for the *ssrA* gene in sputum [57]. *C. pneumoniae* was identified by serology (≥4-fold increase in IgG serum antibody titer using Chlamydia MIF IgG serologic tests [FOCUS Diagnostics, Cypress, CA, USA] between baseline and convalescent samples) or RT-PCR positive for the *argR* gene in sputum [57]. Susceptibility testing was performed by broth microdilution according to the Clinical and Laboratory Standards Institute and the European Committee on Antimicrobial Susceptibility Testing [61,62]. Confirmatory identification and susceptibility testing of isolates, urine antigen testing, serology, and RT-PCR were performed by a central laboratory (Covance Central Laboratory Services, Indianapolis, IN, USA) and specialized laboratories (for RT-PCR of sputum: Accelerō^®^ Bioanalytics GmbH, Berlin, Germany; for all other specialized testing: University of Alabama at Birmingham Diagnostic Mycoplasma Laboratory, Birmingham, AL, USA [*M. pneumoniae]*; Special Pathogens Laboratory, The Legionella Experts^®^, Pittsburgh, PA, USA [*L. pneumophila]*).

### 4.3. Efficacy Assessments

Only patients with baseline atypical pathogens were included in the analyses described herein. Within this patient subgroup, efficacy analyses are presented for the microbiological intent-to-treat (microITT) population (randomized patients with ≥1 baseline CABP-causing pathogen) and the microITT-2 population (randomized patients with ≥1 baseline CABP-causing pathogen detected by a method other than PCR). ECR was assessed at 96 ± 24 h after the first study drug dose. Responders were patients who were alive, showed improvement in ≥2 CABP baseline symptoms, had no worsening of any CABP baseline symptom, and did not receive a nonstudy antibiotic for the treatment of CABP. IACR was assessed at the TOC visit, which was 5–10 days after the last study drug dose. IACR success required resolution or improvement of baseline CABP signs/symptoms such that no additional antibacterial therapy was administered for the current episode of CABP. Microbiological response of success at TOC required either microbiologic eradication (absence of the baseline causative pathogen from repeat cultures obtained between end of treatment [within 2 days after the last study drug dose] and TOC) or presumed eradication (i.e., IACR at TOC was success and culture was not repeated at TOC). TEAEs, defined as any event that started or worsened during or after first study drug dose, were presented for the safety population (all randomized patients who received any amount of study drug) and the microITT population.

### 4.4. Statistical Analyses

For this post hoc pooled analysis, descriptive statistics were generated to characterize patient demographics, baseline clinical characteristics, and efficacy and safety outcomes in the subpopulation of patients with baseline atypical pathogens from LEAP-1 and LEAP-2. These results were interpreted as exploratory descriptive analyses; therefore, no inferential testing was conducted.

## 5. Conclusions

In conclusion, lefamulin was well tolerated and led to high clinical response rates in adults with CABP caused by atypical pathogens, including when given as 5-day oral therapy, regardless of complications such as age or comorbidity. This post hoc analysis suggests that lefamulin may provide a new empiric IV and oral monotherapy alternative to fluoroquinolones and macrolides in patients with CABP caused by atypical pathogens.

## Figures and Tables

**Figure 1 antibiotics-10-01489-f001:**
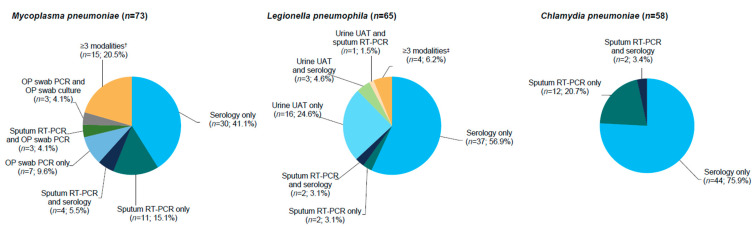
Diagnostic modalities for patients with atypical pathogens detected at baseline * (pooled microITT population [combined treatment groups]); * A patient could have had >1 pathogen identified. Multiple isolates of the same species from the same patient identified by the same testing modality were counted only once. Patients were only counted once for each pathogen based on the unique diagnostic modality or combination of diagnostic modalities by which the pathogen was identified. RT-PCR was performed on OP samples; if RT-PCR was positive for *M. pneumoniae*, OP samples were used for isolation of *M. pneumoniae* and for subsequent susceptibility testing. On some occasions, RT-PCR and culture were performed in parallel. Inclusion of *L. pneumophila* as a baseline pathogen from sputum culture did not require an adequate Gram stain. Culture of *C. pneumoniae* by the local laboratories was allowed per protocol, but it was not cultured successfully by any of the laboratories. ^†^ Includes sputum RT-PCR, serology, and OP swab PCR; sputum RT-PCR, OP swab PCR, and OP swab culture; serology, OP swab PCR, and OP swab culture; and sputum RT-PCR, serology, OP swab PCR, and OP swab culture. ^‡^ Includes urine UAT, sputum RT-PCR, and serology; and sputum culture, urine UAT, sputum RT-PCR, and serology. CABP, community-acquired bacterial pneumonia; microITT, microbiological intent to treat; *n*, number of patients with the respective baseline pathogen; OP, oropharyngeal; PCR, polymerase chain reaction; RT-PCR, real-time PCR; UAT, urine antigen testing.

**Figure 2 antibiotics-10-01489-f002:**
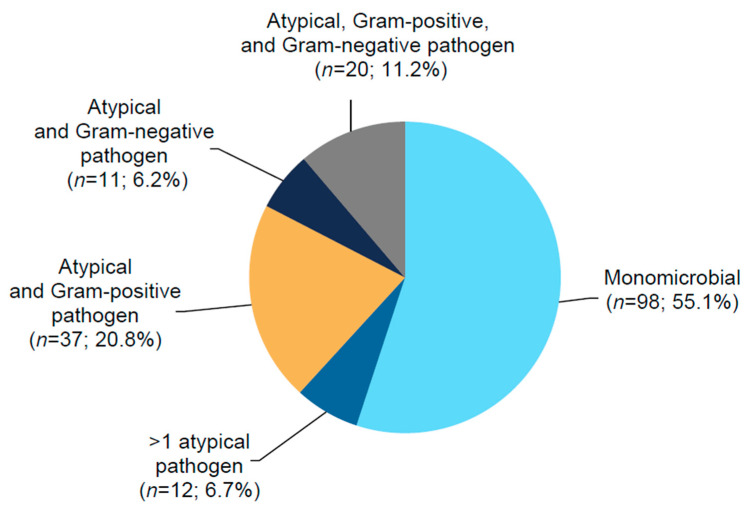
Pathogen distribution for patients with atypical pathogens at baseline * (pooled microITT population [combined treatment groups]); * A patient could have had >1 pathogen identified. Patients were only counted once based on their unique pathogen grouping. CABP, community-acquired bacterial pneumonia; microITT, microbiological intent to treat; *n*, number of patients with the respective baseline pathogen.

**Figure 3 antibiotics-10-01489-f003:**
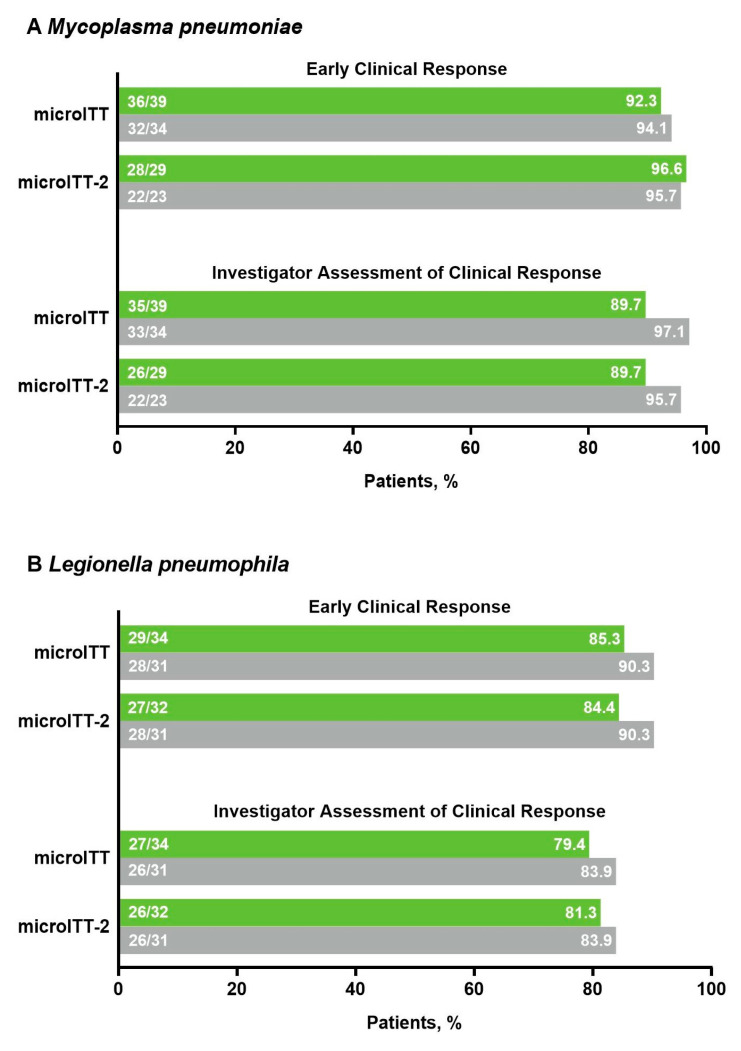
Early clinical response and investigator assessment of clinical response at TOC by analysis population in patients with (**A**) *Mycoplasma pneumoniae*, (**B**) *Legionella pneumophila*, or (**C**) *Chlamydia pneumoniae* at baseline; CABP, community-acquired bacterial pneumonia; microITT, microbiological intent to treat; microITT-2, microbiological intent to treat-2; TOC, test-of-cure visit.

**Figure 4 antibiotics-10-01489-f004:**
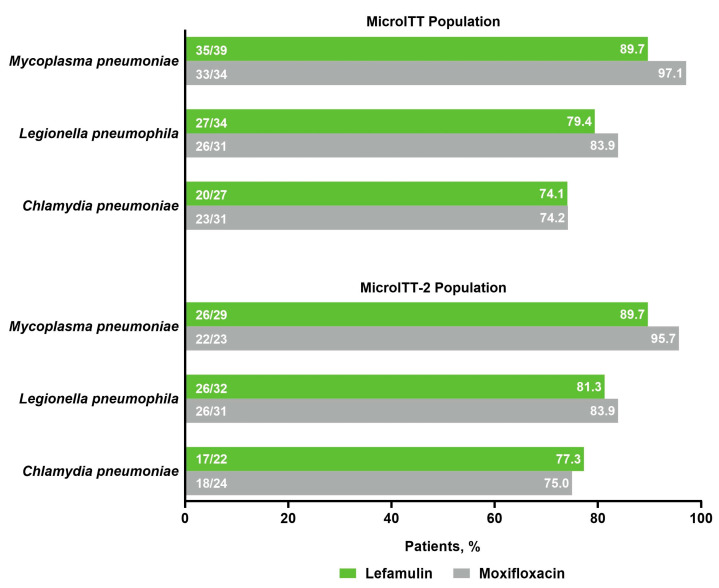
Microbiological response of success at TOC * in patients with atypical pathogens at baseline in the pooled microITT and microITT-2 populations; * Microbiological response of success at TOC was defined as either microbiologic eradication (absence of the baseline causative pathogen from repeat cultures obtained between EOT and TOC) or presumed eradication (the IACR at TOC was success and culture was not repeated at TOC). Note: see Materials and Methods for definitions of response. EOT, end of treatment; IACR, investigator assessment of clinical response; microITT, microbiological intent to treat; microITT-2, microbiological intent to treat-2; TOC, test-of-cure visit.

**Table 1 antibiotics-10-01489-t001:** Patient demographic and baseline characteristics.

Parameter	All Patients (Pooled ITT Population)		Patients with Atypical Pathogens * (Pooled microITT Population)	
	Lefamulin (*n* = 646)	Moxifloxacin (*n* = 643)	Lefamulin (*n* = 91)	Moxifloxacin (*n* = 87)
**Age, y, mean (SD)**	58.9 (16.5)	58.5 (15.7)	54.7 (17.8)	55.6 (17.5)
**Age ≥ 65 y, *n* (%)**	268 (41.5)	249 (38.7)	28 (30.8)	32 (36.8)
**Male, *n* (%)**	377 (58.4)	340 (52.9)	53 (58.2)	49 (56.3)
**White, *n* (%)**	513 (79.4)	509 (79.2)	84 (92.3)	74 (85.1)
**PORT risk class, ^†^ *n* (%)**				
I	1 (0.2)	2 (0.3)	0	0
II	183 (28.3)	190 (29.5)	26 (28.6)	21 (24.1)
III	341 (52.8)	334 (51.9)	49 (53.8)	44 (50.6)
IV	116 (18.0)	112 (17.4)	16 (17.6)	22 (25.3)
V	5 (0.8)	5 (0.8)	0	0
**CURB-65 score, ^‡^ *n* (%)**				
0–2	610 (94.4)	604 (93.9)	87 (95.6)	80 (92.0)
3–5	36 (5.6)	39 (6.1)	4 (4.4)	7 (8.0)
**Minor ATS severity** **criteria, ^§^ *n* (%)**	85 (13.2)	85 (13.2)	15 (16.5)	9 (10.3)
**Modified ATS severity** **criteria, ^||^ *n* (%)**	53 (8.2)	57 (8.9)	8 (8.8)	7 (8.0)
**Multilobar pneumonia, *n* (%)**	170 (26.3)	177 (27.5)	20 (22.0)	17 (19.5)
**SIRS, *n* (%)**	621 (96.1)	609 (94.7)	89 (97.8)	82 (94.3)
**Bacteremic, *n* (%)**	13 (2.0)	12 (1.9)	0	1 (1.1)
**Prior antibiotic use, ^#^ *n* (%)**	147 (22.8)	145 (22.6)	28 (30.8)	23 (26.4)
**Renal status, ^**^ *n* (%)**				
Normal	311 (48.1)	312 (48.5)	56 (61.5)	46 (52.9)
Mild impairment	201 (31.1)	192 (29.9)	25 (27.5)	26 (29.9)
Moderate impairment	125 (19.3)	132 (20.5)	8 (8.8)	15 (17.2)
Severe impairment	7 (1.1)	6 (0.9)	2 (2.2)	0
Missing	2 (0.3)	1 (0.2)	0	0
**Medical history, ^††^ *n* (%)**				
Smoking history	284 (44.0)	242 (37.6)	35 (38.5)	25 (28.7)
Hypertension	248 (38.4)	253 (39.3)	35 (38.5)	29 (33.3)
Asthma/COPD	119 (18.4)	113 (17.6)	10 (11.0)	10 (11.5)
Diabetes mellitus	80 (12.4)	88 (13.7)	7 (7.7)	12 (13.8)
**Baseline pathogen, ^‡‡^ *n* (%)**				
*Mycoplasma pneumoniae*	39 (6.0)	34 (5.3)	39 (42.9)	34 (39.1)
*Legionella pneumophila*	34 (5.3)	31 (4.8)	34 (37.4)	31 (35.6)
*Chlamydia pneumoniae*	27 (4.2)	31 (4.8)	27 (29.7)	31 (35.6)
*Streptococcus pneumoniae*	216 (33.4)	223 (34.7)	24 (26.4)	29 (33.3)
*Haemophilus influenzae*	107 (16.6)	105 (16.3)	8 (8.8)	13 (14.9)
*Moraxella catarrhalis*	46 (7.1)	22 (3.4)	7 (7.7)	1 (1.1)
*Staphylococcus aureus*	23 (3.6)	10 (1.6)	5 (5.5)	1 (1.1)

ATS, American Thoracic Society; BUN, blood urea nitrogen; CABP, community-acquired bacterial pneumonia; COPD, chronic obstructive pulmonary disease; CrCl, creatinine clearance; HLT, high-level term; ITT, intent to treat; MedDRA, *Medical Dictionary for Regulatory Activities*; microITT, microbiological ITT; NEC, not elsewhere classified; PORT, Pneumonia Outcomes Research Team; SIRS, systemic inflammatory response syndrome; WBC, white blood cell (count). * Defined as *M. pneumoniae*, *L. pneumophila*, and *C. pneumoniae*. ^†^ PORT risk class calculated programmatically using site data reported in the electronic case report form was not always consistent with the site-reported PORT risk class used for enrollment/stratification. ^‡^ Defined as confusion of new onset, BUN > 19 mg/dL, respiratory rate ≥ 30 breaths/min, systolic blood pressure < 90 mm Hg or diastolic blood pressure ≤ 60 mm Hg, and age ≥ 65 years. **^§^** Defined as baseline presence of ≥3 of the following nine criteria: respiratory rate ≥ 30 breaths/min, O_2_ saturation < 90% or PaO_2_ < 60 mm Hg, BUN ≥ 20 mg/dL, WBC < 4000 cells/mm^3^, confusion, multilobar infiltrates, platelets < 100,000 cells/mm^3^, temperature < 36 °C, or systolic blood pressure < 90 mm Hg [17]. ^||^ Defined as baseline presence of ≥3 of the following six criteria: respiratory rate ≥ 30 breaths/min, SpO_2_/FiO_2_ < 274 where SpO_2_/FiO_2_ = 64 + 0.84 (PaO_2_/FiO_2_), BUN ≥ 20 mg/dL, confusion, age ≥ 65 years, or multilobar infiltrates [43]. Defined as baseline presence of ≥2 of the following four criteria: temperature < 36 °C or >38 °C; heart rate >90 bpm; respiratory rate > 20 breaths/min; and WBC < 4000 cells/mm^3^, WBC > 12,000 cells/mm^3^, or immature polymorphonuclear neutrophils > 10%. ^#^ Patients received a single dose of short-acting systemic antibacterial medication ≤ 72 h before randomization; randomization was stratified and capped such that ≤25% of the total ITT population met these criteria. ^**^ Defined as normal (CrCl ≥ 90 mL/min), mild (CrCl 60– < 90 mL/min), moderate (CrCl 30– < 60 mL/min), and severe (CrCl < 30 mL/min). ^††^ Medical history terms were defined as follows: hypertension = MedDRA HLT “vascular hypertensive disorders NEC”; asthma/COPD = MedDRA HLT “bronchospasm and obstruction”; diabetes mellitus = MedDRA HLT “diabetes mellitus (incl subtypes)”. ^‡‡^ Among the subpopulation of patients with atypical pathogens (*M. pneumoniae, L. pneumophila, C. pneumoniae*), all patients had ≥1 atypical pathogen at baseline, with the corresponding infections being either mono- or polymicrobial. Within those polymicrobial infections that occurred in patients with atypical pathogens, additional baseline pathogens of *S. pneumoniae*, *H. influenzae*, *M. catarrhalis,* and *S. aureus* were identified.

**Table 2 antibiotics-10-01489-t002:** Overall summary of TEAEs.

Patients, *n* (%)	All Patients (Pooled Safety Population)		Patients with Atypical Pathogens * at Baseline (Pooled microITT Population)	
	Lefamulin (*n* = 641)	Moxifloxacin (*n* = 641)	Lefamulin (*n* = 91)	Moxifloxacin (*n* = 87)
**Any TEAE ^†^**	224 (34.9)	195 (30.4)	31 (34.1)	28 (32.2)
**Mild**	119 (18.6)	117 (18.3)	16 (17.6)	16 (18.4)
**Moderate**	78 (12.2)	55 (8.6)	12 (13.2)	7 (8.0)
**Severe**	27 (4.2)	23 (3.6)	3 (3.3)	5 (5.7)
**Related TEAE ^‡^**	99 (15.4)	68 (10.6)	8 (8.8)	7 (8.0)
**Serious TEAE**	36 (5.6)	31 (4.8)	6 (6.6)	5 (5.7)
**Related serious TEAE**	3 (0.5)	2 (0.3)	0	0
**TEAE leading to study** **drug discontinuation**	20 (3.1)	21 (3.3)	0	4 (4.6)
**TEAE leading to death** **(over entire study duration)**	11 (1.7)	8 (1.2)	1 (1.1) ^§^	0
**28d all-cause mortality—deceased** **at Day 28 ^||^**	8 (1.2)	7 (1.1)	0	0
**TEAEs by SOC in ≥5% of patients in any treatment group**				
Gastrointestinal disorders	84 (13.1)	65 (10.1)	7 (7.7)	7 (8.0)
Infections and infestations	47 (7.3)	40 (6.2)	7 (7.7)	7 (8.0)
Investigations	31 (4.8)	26 (4.1)	5 (5.5)	4 (4.6)
Respiratory, thoracic, and mediastinal disorders	29 (4.5)	28 (4.4)	5 (5.5)	2 (2.3)

COPD, chronic obstructive pulmonary disease; MedDRA, *Medical Dictionary for Regulatory Activities*; microITT, microbiological intent to treat; PORT, Pneumonia Outcomes Research Team; PT, preferred term; SOC, system organ class; TEAE, treatment-emergent adverse event. ***** Defined as *M. pneumoniae*, *L. pneumophila*, and *C. pneumoniae*. ^†^ TEAEs started or worsened during or after first study drug administration (an adverse event with an unknown start date or partial date was categorized as a TEAE); patients with multiple events in a given category were only counted once. ^‡^ TEAEs that were “Definitely”, “Probably”, or “Possibly” related to the study drug. If the TEAE relationship was missing, it was treated as “Related”. ^§^ One patient (aged 70 years; PORT risk class II; moderate renal impairment [creatinine clearance 30 to <60 mL/min] at baseline; history of hypertension and COPD; baseline pathogens *Haemophilus influenzae*, *Haemophilus parainfluenzae*, and *Mycoplasma pneumoniae*) in the lefamulin group had a TEAE leading to death after study day 28; the patient died on study day 271 from acute myeloid leukemia (first reported on study day 269). ^||^ Assessed in the intent-to-treat population (lefamulin *n* = 646; moxifloxacin *n* = 643); details of deaths have been reported elsewhere [40,41]. Although a patient may have had >1 TEAE, the patient was counted only once within an SOC category and once within a PT category. The same patient may have contributed ≥2 PTs in the same SOC category, but the patient was only counted once toward that SOC category. Adverse events were coded using MedDRA version 20.0 (MedDRA MSSO, Herndon, VA, USA).

## Data Availability

The data presented in this study are available on request from the corresponding author.

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
