# Peer review of "Lefamulin in Patients with Community-Acquired Bacterial Pneumonia Caused by Atypical Respiratory Pathogens: Pooled Results from Two Phase 3 Trials"

_antibiotics, 2021, doi:10.3390/antibiotics10121489_

Round 1

Reviewer 1 Report

  1. Abbreviations should not be used in Abstract, but rather be written in full spelling.
  2. Supplementary tables (Table S1 and S2) were not included in the manuscript.
  3. In Table 2, each cause of death in both lefamulin-treated and moxifloxacin-treated arms should be described in full.
  4. In Table 2, under the section of "TEAEs by SOC in >5% of patients in any treatment group", what does "infections and infestations" or "investigations" mean?
  5. For efficacy assessments, the authors analysed a microbiological intent-to treat population who have >1 baseline pathogens (microITT) and another population who have >1 baseline pathogens detected by methods other than PCR (microITT2). What is the purpose of these differentiation?
  6. For statistical analyses, the authors argue that they did not perform inferential statistics because the results of their analyses were interpreted as exploratory descriptive analyses. However, it is assumed that the authors performed randomised trial comparing lefamulin and moxifloxacin, assuming that moxifloxacin is the standard care for community-acquired bacterial pneumonia. If this assumption is correct, the efficacy of lefamulin should be judged by comparing its efficacy with the efficacy of moxifloxacin by inferential statistics. Therefore, the authors' claim that lefamulin was effective against atypical pathogens seems to be unproved. 

Author Response

    1. Abbreviations should not be used in Abstract, but rather be written in full spelling.

Author Response: We have removed all abbreviations from the Abstract apart from acronyms LEAP-1 and LEAP-2, which are the names of the clinical studies.

  1. Supplementary tables (Table S1 and S2) were not included in the manuscript.

Author Response: We confirm that the Supplementary tables were provided at original submission, within a zipped folder with the manuscript as directed by the Instructions to Authors. However, for the Reviewers’ convenience and reference, we have also embedded the Supplementary table file within a comment box on page 12 of the revised manuscript document.

  1. In Table 2, each cause of death in both lefamulin treated and moxifloxacin-treated arms should be described in full.

Author Response: These details have been published previously, in the corresponding primary manuscript for each of the LEAP-1 and LEAP-2 studies (File TM, et al. Clin. Infect. Dis. 2019;69:1856-67 and Alexander E, et al. JAMA 2019;322: 1661-71). As noted in the product label for lefamulin, mortality through day 28 was low (approximately 1%) and similar between treatment arms. Narratives of the deaths are also included in the materials submitted to the US Food and Drug Administration, which are publicly available at https://www.accessdata.fda.gov/drugsatfda_docs/nda/2019/211672Orig1s000,%20211673Orig1s000MultidisciplineR.pdf.

Given these prior reports, inclusion of the full narratives for the deaths among all patients would be outside the scope of the current manuscript, which focuses only on the subset of patients infected with atypical pathogens. However, we have added a footnote in Table 2 indicating where a reader can find details of all patient deaths.

  1. In Table 2, under the section of "TEAEs by SOC in >5% of patients in any treatment group", what does "infections and infestations" or "investigations" mean?

Author Response: Clinical trials are often designed in partnership with regulatory authorities so that the results can be used as the basis for regulatory approval. Because of this, one of the standard conventions for such clinical studies is to classify adverse events using the standard terminology and hierarchical structure of the Medical Dictionary for Regulatory Activities (MedDRA). The highest level of the MedDRA hierarchy is the System Organ Class (SOC), and more than 80,000 terms for individual medical events are categorized into 27 SOCs based on etiology (eg, the SOC of Infections and Infestations), manifestation site (eg, the SOC of Gastrointestinal Disorders), or purpose (eg, the SOC of Surgical and Medical Procedures).

The four SOCs included in Table 2 (Gastrointestinal Disorders; Infections and Infestations; Investigations; and Respiratory, Thoracic and Mediastinal Disorders) are the standard MedDRA SOCs that collectively included the most frequently reported (ie, >5% of patients in any treatment arm) treatment-emergent adverse events from these trials. The SOC of Gastrointestinal Disorders, for example, included adverse events such as diarrhea, nausea, and vomiting.

  1. For efficacy assessments, the authors analysed a microbiological intent-to treat population who have >1 baseline pathogens (microITT) and another population who have >1 baseline pathogens detected by methods other than PCR (microITT2). What is the purpose of these differentiation?

Author Response: Regulatory authorities often require multiple analysis populations, as was the case for the two registration trials included in this post hoc analysis. In this case, the rigorous enrollment criteria for the trials requiring all patients to have clinical signs and symptoms of pneumonia were accompanied by multiple diagnostic methods, including PCR. Although the PCR diagnostic methods for bacterial detection were validated before use in the LEAP-1 and LEAP-2 trials and conservative cut-off values for the DNA and organism concentrations were set, regulatory authorities requested the analysis in the microITT-2 population because of the lack of causality between infection and detection of the organism by real-time PCR. The microITT-2 population was identical to the microITT population (ie, all randomized treated patients with an identified baseline pathogen), except that the microITT-2 population excluded patients whose baseline pathogens were identified only using PCR methods. As noted in the Results text (page 4, first paragraph) and shown in Figure 2, most of the atypical pathogens in these studies (71%–97%) were identified by at least 1 standard diagnostic modality (culture, serology, or urinary antigen test) and would have been included in both analysis populations. We have added the following additional context for the two populations in the Discussion section (page 10, paragraph 3):

The use of PCR-based diagnostic modalities has the potential to identify pathogens that are not etiologically or clinically relevant to a patient’s diagnosis. However, our results indicate that clinical response rates were high and similar between treatment groups regardless of whether the analysis population included (microITT population) or excluded (microITT-2 population) patients with baseline pathogens identified using PCR only.

  1. For statistical analyses, the authors argue that they did not perform inferential statistics because the results of their analyses were interpreted as exploratory descriptive analyses. However, it is assumed that the authors performed randomised trial comparing lefamulin and moxifloxacin, assuming that moxifloxacin is the standard care for community-acquired bacterial pneumonia. If this assumption is correct, the efficacy of lefamulin should be judged by comparing its efficacy with the efficacy of moxifloxacin by inferential statistics. Therefore, the authors' claim that lefamulin was effective against atypical pathogens seems to be unproved

Author Response: This manuscript reports a post hoc analysis of pooled data from two trials, both of which were designed to test noninferiority of lefamulin versus moxifloxacin. For each of these two trials, the a priori analyses were designed to have sufficient power to establish non-inferiority of lefamulin to moxifloxacin using a 10% margin for the ECR and IACR outcomes. This non-inferiority margin was considered statistically valid only for the pre-specified analyses reported individually for each study (File TM, et al. Clin. Infect. Dis. 2019, 69, 1856-1867 and Alexander E, et al. JAMA 2019, 322, 1661-1671) and for a pre-specified pooled analysis reported elsewhere (File TM, et al. BMC Pulm. Med. 2021, 21, 154). Post hoc analyses such as those reported herein, which were designed following completion of the pre-specified analyses, are subject to bias. Therefore, such analyses are not suitable for drawing statistically significant conclusions and should be used only to conduct exploratory hypothesis testing.

Given these considerations, we cannot make the conclusion that lefamulin is noninferior to moxifloxacin against atypical pathogens. We have edited the language throughout the manuscript to avoid the implication that this post hoc analysis can be used to establish non-inferiority. Rather, we define the observed clinical response rates (74.1% to 97.1%) as ‘high’ and draw the much less statistically robust conclusion that these response rates were high with both study drugs. No statistical comparisons between lefamulin and moxifloxacin were made, and no statistical conclusions regarding any such comparison were drawn. The limitations section of this manuscript (page 10, paragraph 3) cautions readers against making such conclusions or comparisons.

Reviewer 2 Report

In their manuscript, Paukner et al. re-analysed the data of the LEAP-1 and the LEAP-2 trial with regard to atypical pathogens. The LEAP trials were phase 3 clinical studies comparing lefamulin versus moxifloxacin for community-acquired bacterial pneumonia (CABP). Here they compared the treatment outcomes for patients with positive microbiological diagnostics for atypical pathogens and found good early clinical response rates (ECR) as well as investigator assessment of clinical response rates (IACR) for the lefamulin group and the moxifloxacin group without any statistically significant differences. There is definitely a need for new antimicrobial substances targeting atypical pathogens in pneumonia. Therefore, the idea for this analysis is good and novel.

However, the evaluation is – as it is widely known – very difficult. The microbiological diagnostics for atypical bacteria in pneumonia (M. pneumonia, C. pneumonia or L. pneumonia) is difficult and frequently indecisive. This is a problem for the present analysis as well. In the majority of cases the diagnostics were made by serology only (for L. pneumonia this was the case in 75% of patients – although there is the urine antigen test for this pathogen as well which was probably negative). Additionally, up to 50% of patients had also evidence of a typical bacterium (mostly S. pneumonia). Taken together, it is not clear if the pneumonia in the patients studied here was indeed caused by an atypical pathogen, at least not for all of them.

As the authors state themselves in the “Discussion” section the number of patients studied is small and was not powered to detect statistically significant differences.

Therefore, the authors should change the manuscript accordingly and make these concerns clear throughout the entire manuscript. In addition, they should discuss the point that the severity of the disease in most patients was mild to moderate (CURB-65 score 0 – 2). So their conclusions hold true for these cases only.

Author Response

  1. In their manuscript, Paukner et al. re-analysed the data of the LEAP-1 and the LEAP-2 trial with regard to atypical pathogens. The LEAP trials were phase 3 clinical studies comparing lefamulin versus moxifloxacin for community-acquired bacterial pneumonia (CABP). Here they compared the treatment outcomes for patients with positive microbiological diagnostics for atypical pathogens and found good early clinical response rates (ECR) as well as investigator assessment of clinical response rates (IACR) for the lefamulin group and the moxifloxacin group without any statistically significant differences. There is definitely a need for new antimicrobial substances targeting atypical pathogens in pneumonia. Therefore, the idea for this analysis is good and novel.

Author Response: We thank the Reviewer for the comments and have revised the manuscript accordingly as noted below.

  1. However, the evaluation is – as it is widely known – very difficult. The microbiological diagnostics for atypical bacteria in pneumonia ( pneumoniae, C. pneumoniae or L. pneumophila) is difficult and frequently indecisive. This is a problem for the present analysis as well. In the majority of cases the diagnostics were made by serology only (for L. pneumophila this was the case in 75% of patients – although there is the urine antigen test for this pathogen as well which was probably negative). Additionally, up to 50% of patients had also evidence of a typical bacterium (mostly S. pneumoniae). Taken together, it is not clear if the pneumonia in the patients studied here was indeed caused by an atypical pathogen, at least not for all of them.

As the authors state themselves in the “Discussion” section the number of patients studied is small and was not powered to detect statistically significant differences. Therefore, the authors should change the manuscript accordingly and make these concerns clear throughout the entire manuscript.

Author Response: We agree with the Reviewer that this analysis was limited by the small patient numbers and the possibility that an atypical pathogen may not have been the causative pathogen in all patients. The following text has been added to the Discussion section (page 10, paragraph 3):

The use of PCR-based diagnostic modalities has the potential to identify pathogens that are not etiologically or clinically relevant to a patient’s diagnosis. However, our results indicate that clinical response rates were high and similar between treatment groups regardless of whether the analysis population included (microITT population) or excluded (microITT-2 population) patients with baseline pathogens identified using PCR only.

We have also edited the language throughout the text to avoid the implication that this post hoc exploratory analysis was designed to make statistical comparisons between the two treatment groups.

  1. In addition, they should discuss the point that the severity of the disease in most patients was mild to moderate (CURB-65 score 0 – 2). So their conclusions hold true for these cases only.

Author Response: We agree with the reviewer that results from clinical trials such as this are not always generalizable to what is seen in clinical practice. We have strengthened the limitation in the Discussion section with regards to disease severity accordingly (page 10, paragraph 3; reproduced below). However, current pneumonia treatment guidelines (Metlay et al. Am J Resp Crit Care Med. 2019;200:e45.) recommend using PORT risk class over CURB-65 scores for prognosis, as PORT risk class “identifies larger proportions of patients as low risk and has a higher discriminative power in predicting mortality.” Most patients in this study had a PORT risk class of III (ie, moderately severe) or greater and approximately 25% had multilobar disease (indicative of more severe disease); therefore, patients with severe pneumonia may have been reasonably represented by the study population.

Finally, these findings may not be generalizable to all patients with CABP caused by atypical pathogens, as the enrollment criteria for the LEAP 1 and LEAP 2 trials may have excluded some patients who would typically be seen in clinical practice. Most patients had CURB-65 scores of 0-2, reflective of mild to moderate disease, potentially limiting generalizability of the results to patients with more severe disease. However, approximately two-thirds of the patients had a PORT risk class of III or greater, and one-quarter had multilobar pneumonia, suggesting that patients with severe pneumonia may have been reasonably represented by the study population.

Round 2

Reviewer 1 Report

The authors have addressed the concerns raised by the reviewer appropriately. The manuscript shall be accepted in the current version.

Author Response

Thank you for your review.

Reviewer 2 Report

The authors have changed their manuscript according to our suggestions in the Discussion section only. We find it neccessary to change it throughout the manuscript (e.g. Results section) and especially to mention the points in the Abstract also. At least the hint to mild to moderate cases should be included in the Abstract - as many readers look at the Abstract only.

Additionally, they have discussed PCR testing in the Discussion section. Our main concern, however, were the cases with the diagnosis based on serology.

Author Response

  1. The authors have changed their manuscript according to our suggestions in the Discussion section only. We find it necessary to change it throughout the manuscript (e.g. Results section) and especially to mention the points in the Abstract also. At least the hint to mild to moderate cases should be included in the Abstract - as many readers look at the Abstract only.

Author Response: The following sentence describing patients’ disease severity has been added to the Abstract (page 1) and the Results section (page 3):

In terms of disease severity, more than 90% of patients had CURB-65 (confusion of new onset, blood urea nitrogen >19 mg/dL, respiratory rate ≥30 breaths/min, blood pressure <90 mm Hg systolic or ≤60 mm Hg diastolic, and age ≥65 years) scores of 0‒2 (Table 1); approximately 50% of patients had PORT (Pneumonia Outcomes Research Team) risk class of III, and the remaining patients were more likely to have PORT risk class of II or IV versus V.

For additional clarity and completeness, Table 1 has been revised to show the full breakdown of individual PORT risk class scores (ie, I, II, III, IV, V instead of I/II, III, IV/V) Because the LEAP-1 and LEAP-2 studies were designed to enroll patients with PORT scores of III‒V and II‒IV, respectively, the number of patients with PORT risk class of I/II in LEAP-1 and PORT risk class of I in LEAP-2 was minimal.

The following sentence summarizing study limitations has also been added to the Abstract (page 1):

Limitations to this analysis include: its post hoc nature, the small numbers of patients infected with atypical pathogens, the possibility of PCR-based diagnostic methods to identify non-etiologically relevant pathogens, and the possibility that these findings may not be generalizable to all patients.

  1. Additionally, they have discussed PCR testing in the Discussion section. Our main concern, however, were the cases with the diagnosis based on serology.

Author Response: The observation that serology was the most common diagnostic modality has been added to the Abstract (page 1) and the Results section (page 4). Serology, and specifically the ≥4-fold IgG increase at convalescence compared with baseline (and not just high IgG or IgM titers), is an accepted methodology for baseline pathogen identification by health authorities, including the US FDA and EMA. In the case of atypical pathogens, serology is an important method to detect these organisms, since cultivation can be very challenging or impossible. Given that a ≥4-fold increase in antibody response is an accepted and valid method for diagnosis, we do not consider diagnosis via serology to be a limitation of the study.